**Subject Area:**
biochemistry/cellular biology

mitochondria, mitochondrial biogenesis, metabolism

**Author for correspondence:**
Diana Stojanovski
e-mail: d.stojanovski@unimelb.edu.au

# Mitochondria—hubs for regulating cellular biochemistry: emerging concepts and networks

Alexander J. Anderson, Thomas D. Jackson, David A. Stroud and Diana Stojanovski

Department of Biochemistry and Molecular Biology and The Bio21 Molecular Science and Biotechnology Institute, The University of Melbourne, Parkville, Victoria, 3010, Australia

DS, 0000-0002-0199-3222

Mitochondria are iconic structures in biochemistry and cell biology, traditionally referred to as the powerhouse of the cell due to a central role in energy production. However, modern-day mitochondria are recognized as key players in eukaryotic cell biology and are known to regulate crucial cellular processes, including calcium signalling, cell metabolism and cell death, to name a few. In this review, we will discuss foundational knowledge in mitochondrial biology and provide snapshots of recent advances that showcase how mitochondrial function regulates other cellular responses.

## 1. Introduction

All modern-day eukaryotes are believed to have arisen from a primordial ancestor that engulfed an α-protobacterium with the capacity for respiration [1]. This event gave rise to modern-day mitochondria, an event that is now deeply integrated in eukaryotic cell homeostasis and survival. Mitochondria are dynamic networks capable of remodelling their morphology and activity. They provide energy and biomolecules for the cell, in addition to contributing to pathways of cell stress, immune responses, intra- and intercellular signalling, cell-cycle control and cell death. The unique biology of mitochondria underpins their influence on the cell and the ability to calibrate their structure and proteome is an efficacious means of adapting their function. As such, we will begin with a brief outline of three fundamental concepts in mitochondrial biology: (i) mitochondrial ultrastructure; (ii) mitochondrial protein import; and (iii) mitochondrial dynamics. This will inform subsequent discussion of mitochondria as key players in broad and diverse roles, including metabolism, signal transduction, immunity, cell cycle, cell differentiation, cell death and stress.

## 2. Mitochondrial ultrastructure, dynamics and protein import

### 2.1. Mitochondrial ultrastructure

Mitochondria have a double membrane that defines four compartments: the outer membrane, the intermembrane space, the inner membrane and the matrix. The architecture of the inner membrane is malleable and typically convoluted into folded invaginations, called cristae, that dictate the spatial arrangement of proteins [2]. Remodelling cristae structure of cristae can also alter enzymatic flux between the compartments, consistent with the diverse cristae structures observed across cell types with different metabolic demands [2]. The recently described MICOS complex (mitochondrial contact site and cristae organizing system) is required to maintain cristae morphology [3] (figure 1). Loss of MICOS assembly

royalsocietypublishing.org/journal/rsob    Open Biol. 9: 190126

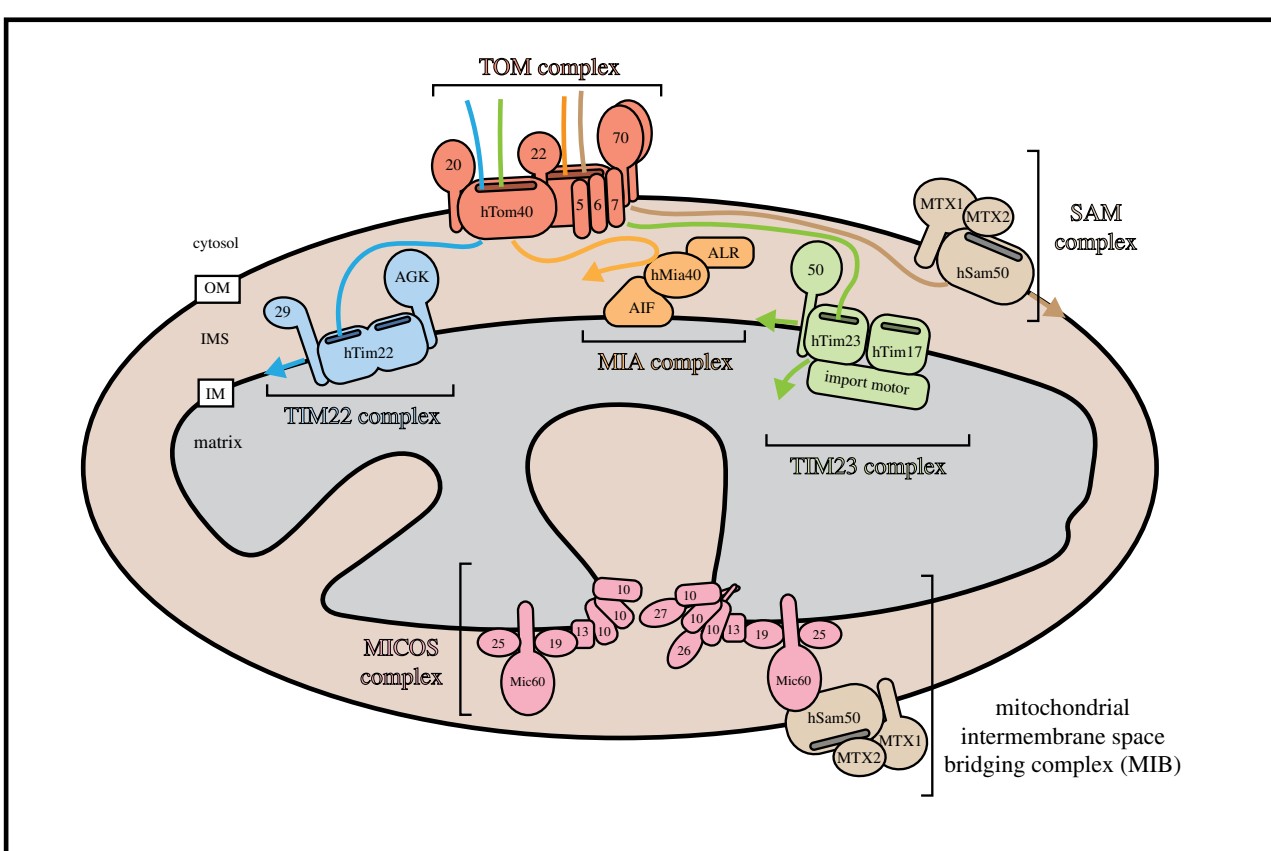

**Figure 1.** Nuclear-encoded mitochondrial proteins are imported by multi-subunit translocases. Mitochondrial proteins synthesized in the cytosol are imported into mitochondria post-translationally. The TOM complex at the outer membrane serves as a general protein entry gate. hTom40 forms the pore of the translocase, while hTom20, hTom22 and hTom70 function as receptors. hTom22 plays an additional role in the assembly of the complex. hTom5, hTom6 and hTom7, collectively called the small TOMs, regulate the dynamics and assembly of the complex. The TIM22 complex at the inner membrane mediates the import of multi-pass transmembrane proteins into the inner membrane. hTim22 forms the pore through which proteins are inserted, while AGK and hTim29 function as receptors and in complex assembly. The TIM23 complex can translocate precursor proteins into the matrix or the inner membrane. hTim23 and hTim17 form the channel pore, and hTim50 functions as a receptor for precursors. The core complex associates with an import motor that helps to translocate proteins into the matrix in an ATP-dependent manner. The MIA complex mediates the import of soluble intermembrane space proteins by catalysing the formation of disulfide bonds. hMia40 carries out the disulfide bond formation and is anchored to the inner membrane through an interaction with AIF. ALR removes electrons from hMia40 so that it can undergo further rounds of catalysis. The SAM complex of the outer membrane mediates insertion of β-barrel proteins into the outer membrane. hSam50 associates with MTX1 and MTX2. Cristae, the large invaginations of the inner mitochondrial membrane, are stabilized by a multi-subunit complex called MICOS. Mic60 is the core subunit of MICOS, which additionally contains Mic10, Mic13, Mic14, Mic19, Mic25, Mic26 and Mic27. MICOS also associates with the SAM complex at the outer membrane to form a structure known as the mitochondrial intermembrane space bridging complex (MIB).

ablates cristae junctions and manifests severe defects in energy metabolism, calcium handling and lipid trafficking [4]. However, it remains unclear how MICOS is regulated by cellular conditions to produce diverse cristae morphologies. Interestingly, disruption of MICOS alters the activity and/or abundance of mitochondrial morphology proteins [5,6]. Perturbations to organelle function have long been associated with gross morphological changes in the mitochondrial network, therefore cristae reorganization by MICOS assembly/disassembly may be an intermediary between function and dynamics. Recently identified associations between MICOS and protein import complexes point to the broad influence of MICOS on mitochondrial function [7,8].

(Recommended further reading on cristae, MICOS and ultrastructure: [2,9,10].)

## 2.2. Mitochondrial protein import

From their endosymbiont origins, human mitochondria have retained only 37 genes in a small circular genome known as the mitochondrial DNA (mtDNA), which encodes 13 polypeptides, 22 tRNAs and 2 rRNAs. The remaining 1000–1500

mitochondrial proteins are nuclear encoded and must be imported and sorted to the relevant mitochondrial compartment following synthesis in the cytosol. Fundamentally, mitochondrial protein import is mediated by multimeric protein complexes known as translocases, which are located at mitochondria (figure 1). Briefly, two major translocases reside in the outer membrane of mitochondria: the Translocase of the Outer Membrane (TOM) complex and the Sorting and Assembly Machinery (SAM). The TOM complex is the initial point of contact for almost all mitochondrial precursors and provides a means of entry into the organelle. Following translocation through TOM, precursor import pathways diverge based on their targeting information and ultimate location within the organelle. β-barrel proteins of the outer membrane are sorted to the SAM complex for integration into the membrane. There are two translocases embedded in the inner membrane of mitochondria: the Translocase of the Inner Membrane (TIM) 22 and 23 (TIM22 and TIM23) complexes. TIM22 mediates the insertion of non-cleavable polytopic membrane proteins into the inner membrane, while the TIM23 complex is responsible for importing precursors across the inner membrane into the matrix or in some instances can laterally release

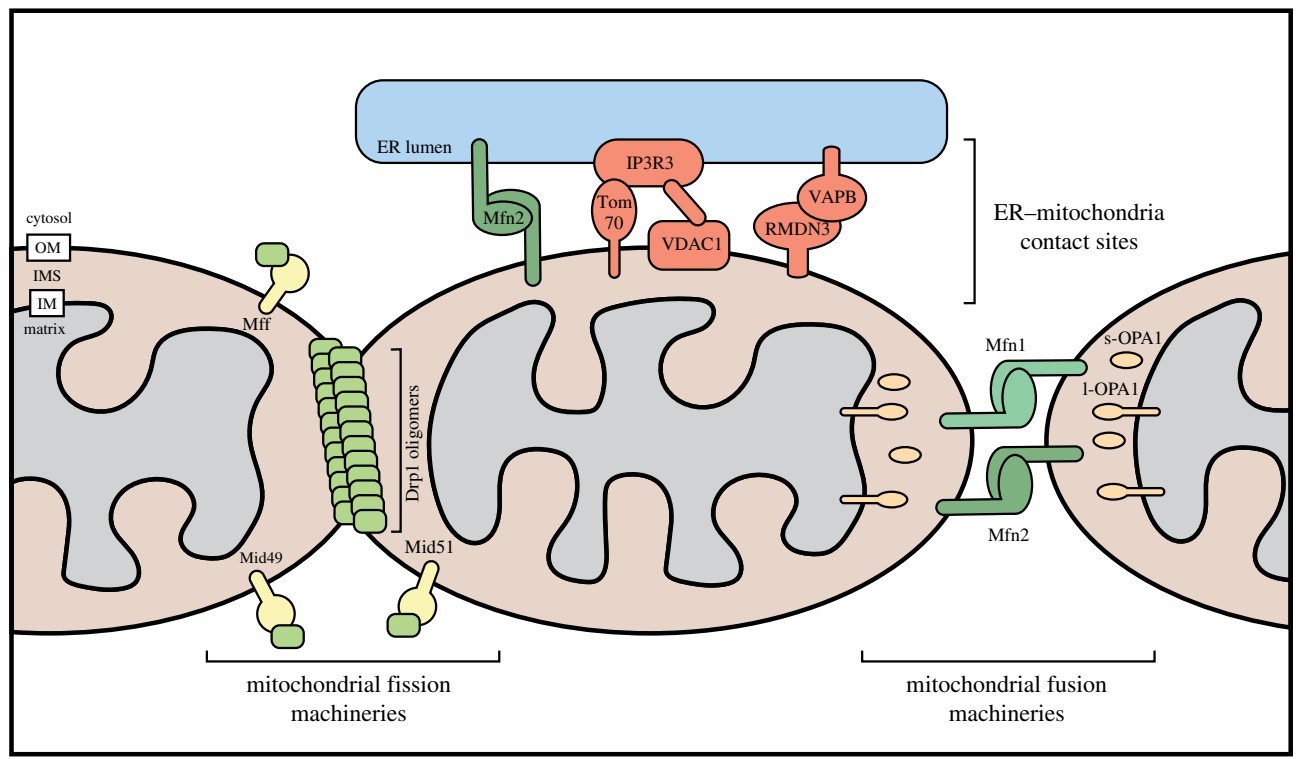

**Figure 2.** Cellular machineries mediating mitochondrial fission, fusion and formation of contact sites with the endoplasmic reticulum. Mitochondria continuously undergo fission and fusion. Fission is mediated by the GTPase Drp1, which can be recruited to the outer mitochondrial membrane by a variety of receptors, including Mff, Fis1, Mid49 and Mid51. Drp1 at the outer membrane can oligomerize into fibrils that constrict mitochondria to initiate fission. Mitochondrial fusion is initiated by tethering of mitochondria through homotypic interactions between Mfn1 and Mfn2 on opposing mitochondria. Inner membrane fusion is mediated by OPA1, which exists as long and short forms generated through proteolysis. Contact sites between the mitochondria and the endoplasmic reticulum (ER) are established and maintained through protein–protein interactions. Interactions occur between Mfn2 molecules on the ER membrane and the outer mitochondrial membrane, and between VAPB on the ER membrane and RMDN3 on the mitochondrial outer membrane. Interactions also occur between IP3R3, a calcium channel on the ER membrane, and VDAC1 and hTom70 on the mitochondrial outer membrane.

transmembrane precursors into the inner membrane. Finally, the Mitochondrial Intermembrane space Assembly (MIA) machinery mediates the import of small cysteine-rich intermembrane space proteins and couples their import to their oxidation [11]. These import pathways and machines have been predominately characterized in fungal organisms; however, in more recent years, analysis in higher eukaryotes has uncovered important physiological consequences due to perturbations in protein import. Specifically, mutations in genes encoding protein import subunits cause distinct mitochondrial diseases with phenotypes ranging from severe muscular defects to neurodegeneration and congenital growth defects [12].

(Recommended further reading on mitochondrial protein import: [13–15].)

## 2.3. Mitochondrial dynamics: fission, fusion and organelle contact sites

As an organellar network, mitochondria undergo fission and fusion to replicate, be recycled, and alter their bioenergetics. Fusion of the outer membrane is mediated by homotypic interactions between the GTPases Mfn1 and Mfn2 on adjacent mitochondria (figure 2) [16], but the domains involved and stepwise mechanism of fusion are still debated. Fusion of the inner membrane is controlled by Opa1, which exists as five isoforms generated by mRNA splicing and proteolytic cleavage (figure 2) [17]. It is believed that the stoichiometry

of these isoforms governs Opa1 interactions with the mitochondrial-specific lipid cardiolipin and, subsequently, fusion events [18]. Mitochondrial fusion is associated with increased ATP production by oxidative phosphorylation and protects against oxidative and proteostatic stress [19]. Conversely, mitochondrial fission is concomitant with a reliance on glycolysis and precedes mitochondrial turnover. Fission is largely dependent on the dynamin-related and cytosolic protein Drp1, which oligomerizes around and constricts mitochondrial tubules (figure 2). The recruitment of Drp1 from the cytosol requires adaptor proteins on the mitochondrial outer membrane, including Mff, Mid49 and Mid51 [20,21], although human Fis1 can promote Drp1-independent mitochondrial fragmentation through inhibition of fusion proteins [22] (figure 2). While conflicting models of Drp1 recruitment have been proposed, its localization and activity are known to be regulated by numerous post-translational modifications [23]. The scission ability of Drp1 oligomers is sterically limited to tubules up to 250 nm diameter, indicating pre-constriction is required for larger mitochondria [24]. This is achieved by the endoplasmic reticulum (ER), which wraps around and constrict tubules to mark future fission sites and aid correct partitioning of mitochondrial contents [25,26].

Mitochondria also engage in extensive dynamic inter-organelle contacts that coordinate functional exchanges between mitochondria and other cellular components [27]. In particular, ER–mitochondria contact sites (ERMCs) facilitate a multitude of functions including mitochondrial fission, coenzyme Q biosynthesis, lipid transfer, $Ca^{2+}$ transfer, mtDNA

royalsocietypublishing.org/journal/rsob    Open Biol. 9: 190126

replication and autophagy [25,26,28–31]. The ER–mitochondria encounter structure (ERMES) has been well characterized in *Saccharomyces cerevisiae* [32], however no human equivalent has been identified [33]. Preliminary work in humans suggests that metazoan ERMCs are tethered by interactions between hTom70 and IP3R3, VDAC1 and IP3R3, RMDN3 and VAPB, Mfn2 homodimers, Vps13a and Pdzd8 with an unknown partner (figure 2) [34–39]. Furthermore, acetylated microtubule 'tracks' have been proposed to maintain these contacts despite the movements and remodelling of the two organellar networks [40]. Other inter-organelle contacts have been described between mitochondria and Golgi [27,41], peroxisomes [42], lysosomes [43], lipid droplets [44] and the plasma membrane [45]. The interconnectivity of mitochondria with cellular components enables significant interplay across various pathways, examples of which we will highlight throughout this review (table 1).

(Recommended further reading on mitochondrial dynamics: [46–48]; on organelle contacts: [49–51].)

## 3. Mitochondria and metabolism

Mitochondria are well known for providing energy to the cell, predominantly by coupling the tricarboxylic acid (TCA) cycle with oxidative phosphorylation. The TCA cycle is a series of eight enzymatic reactions that occur in the matrix to harvest electrons from citrate and its catabolic intermediates (figure 3*a*). The typical input to the cycle is acetyl-CoA, which can be produced from glucose (via glycolysis), fatty acids (via β-oxidation) and amino acids (via deamination) (figure 3*a*). The electrons scavenged throughout the cycle are transferred by NADH and $FADH_2$ to the complexes of the electron transport chain. Complexes I–IV of the electron transport chain shuttle electrons, using their energy to pump protons into the intermembrane space and establish an electrochemical gradient across the inner membrane. Complex V (ATP synthase) releases the protons back into the matrix, using the energy of the electrochemical gradient to produce ATP, the cell's energy currency (figure 3*a*) [52]. Although normally efficient, oxidative phosphorylation is negatively regulated by the accumulation of its toxic by-product, reactive oxygen species (ROS). If unchecked, ROS can cause damage to mitochondria, induce protein aggregation and introduce mutations in DNA [53–55]. Recent advances in cryoelectron microscopy have revealed Complexes I, III and IV can assemble to form supercomplexes thought to reduce the amount of ROS produced during electron transport, as well as enhance respiration rates [56].

Mitochondria also produce fatty acids, amino acids, nucleotides and haem groups for the cell through biosynthetic pathways [57–59]. One such process, one-carbon (1C) metabolism, produces glycine, methionine, nucleotides, phosphatidylcholine and 1C units (methyl-like groups) from serine catabolism through the redox chemistry of folate and its derivatives (figure 3*b*) [60]. These 1C units charge the universal methyl donor *S*-adenosylmethionine required for the methylation of proteins and chromatin [61]. There is now significant evidence of metabolic enzymes and metabolites altering gene expression as reporters of environmental conditions (nutrient availability, hypoxia, oxidative stress) or mitochondrial dysfunction. This has been shown for acetyl-CoA, TCA intermediates, ketones,

lactate, fatty acids and amino acids [62–68]. Emerging studies also indicate cellular nutrient and energy sensing by mTOR kinase regulates mitochondrial biogenesis and protein synthesis [69]. Through downstream effectors of transcription and translation, mTORC1 stimulates mitochondrial biogenesis and oxidative metabolism to meet the energy demand of anabolism [70–72]. Interestingly, the tumour suppressor p53 inhibits mTOR-mediated growth and proliferation to prevent oncogenesis [73,74]. p53 activity increases electron transport chain efficacy [75], mtDNA stability [76,77] and reduced glutathione (GSH) levels [78] to limit ROS production as well as inhibiting glycolysis [79,80], which contributes to the replicative potential of tumour cells [79,81,82]. Thus, metabolism is intimately integrated with other cellular pathways, but is not the sole contribution of mitochondria to signalling mechanisms.

(Recommended further reading on metabolism: [60,83]; on metabolite signalling: [68,84]; on mTOR/p53: [85,86].)

## 4. Signalling

### 4.1. Mitochondria control calcium homeostasis

Calcium ions are common to diverse signalling pathways. The outer mitochondrial membrane is permeable to $Ca^{2+}$, in part due to channel-forming VDAC proteins [87] and export via SLC8A3 [88]. The mitochondrial inner membrane calcium uniporter (MCU) complex regulates transport into the matrix (figure 3*c*). Permeability of the MCU complex is calibrated by two regulatory subunits, MICU1 and MICU2, that are linked by an intermolecular disulfide bond introduced by hMia40 [89,90]. The ability of mitochondria to accumulate $Ca^{2+}$ up to 20-fold higher concentrations than the cytosol allows them to function as buffering systems and re-establish homeostasis following $Ca^{2+}$ bursts [91,92]. Bursts of $Ca^{2+}$ into the cytosol, from across the plasma membrane or intracellular stores, can initiate neurotransmitter release, muscle fibre contraction and transcriptional regulation. In neurons, mitochondrial $Ca^{2+}$ buffering modulates both the propensity and duration of neurotransmitter release [93,94]. In cardiac muscle, contraction is coupled to enhanced mitochondrial ATP production via $Ca^{2+}$-increased activities of TCA cycle enzymes, Complex V and the ADP/ATP transporter [95–98]; an effect maximized by local $Ca^{2+}$ concentrations at ERMCs [29,99] (figure 3*c*). Additionally, mitochondrial $Ca^{2+}$ regulation influences hormone secretion [100], tissue regeneration [101] and interferon-β signalling via the mitochondrial antiviral signalling protein, MAVS [102].

(Recommended further reading on mitochondrial $Ca^{2+}$ signalling: [92,103,104].)

### 4.2. Roles of mitochondria in immune responses

The contribution of mitochondria to immune responses is a growing area of research. Cell-autonomous immune signalling is driven by MAVS at the outer membrane, which acts as a relay point for immune signal transduction. Rig-like receptors in the cytosol undergo conformational changes upon detecting viral RNA or DNA and are recruited to MAVS, particularly at ERMCs [105]. MAVS then dimerizes to enable the binding of multiple downstream signalling adaptors including TRADD, TRAF3 and STING to activate NF-κB and IRF-3/7

**Table 1.** Full names and identifiers of proteins discussed in this review.

| section | protein name | | | | |
| | abbreviation | full name(s) | gene | accession (NCBI; UniProt) | function(s) |
| --- | --- | --- | --- | --- | --- |
| mitochondrial dynamics | Mfn1 | Mitofusin 1 | MFN1 | 55669; Q8IWA4 | outer membrane fusion |
| | Mfn2 | Mitofusin 2 | MFN2 | 9927; O95140 | outer membrane fusion; ER–mitochondria contact |
| | Opa1 | OPA1 mitochondrial dynamin-like GTPase | OPA1 | 4976; O60313 | inner membrane fusion |
| | Drp1 | dynamin-1-like protein; Drp1 | DNM1L | 10059; O00429 | mitochondrial fission |
| | Fis1 | mitochondrial fission protein 1 | FIS1 | 51024; Q9Y3D6 | mitochondrial fission |
| | Mff | mitochondrial fission factor | MFF | 56947; Q9GZY8 | mitochondrial fission |
| | Mid51 | mitochondrial dynamics protein 51 | MIEF1 | 54471; L0R8F8 | mitochondrial fission |
| | Mid49 | mitochondrial dynamics protein 49 | MIEF2 | 125170; Q96C03 | mitochondrial fission |
| organelle contact site | hTom70 | translocase of the outer membrane 70 | TOMM70 | 9868; O94826 | protein import; ER–mitochondria contact |
| | VDAC1 | voltage-dependent anion channel 1 | VDAC1 | 7416; P21796 | ER–mitochondria contact; ion permeability |
| | IP3R3 | inositol 1,4,5-trisphosphate receptor type 3 | ITPR3 | 3710; Q14573 | ER contact sites; calcium transport |
| | RMDN3 | regulator of microtubule dynamics protein 3 | RMDN3 | 55177; Q96TC7 | ER contact sites; calcium transport |
| | VAPB | VAMP associated protein B and C | VAPB | 9217; O95292 | ER contact sites |
| | Vps13a | vacuolar protein sorting 13 homolog A | VPS13A | 23230; Q96RL7 | ER contact sites; lipid transfer |
| | Pdzd8 | PDZ containing 8 | PDZD8 | 118987; Q8NEN9 | ER contact sites; calcium transport |
| metabolism | mTOR | mechanistic target of rapamycin kinase; serine/threonine protein kinase mammalian target of rapamycin | MTOR | 2475; P42345 | metabolic regulation; cell growth |
| | p53 | tumour protein 53 | TP53 | 7157; P04637 | metabolic regulation; cell survival |
| calcium homeostasis | SLC8A3 | solute carrier family 8 member A3 | SLC8A3 | 6547; P57103 | calcium transport |
| | MCU | mitochondrial calcium uniporter | MCU | 90550; Q8NE86 | calcium transport |
| | MICU1 | mitochondrial calcium uptake 1 | MICU1 | 10367; Q9BPX6 | calcium transport, regulation |
| | MICU2 | mitochondrial calcium uptake 2 | MICU2 | 221154; Q8IYU8 | calcium transport, regulation |
| | hMia40 | coiled-coil–helix–coiled-coil–helix domain containing 4; mitochondrial intermembrane space import and assembly 40 homolog | CHCHD4 | 131474; Q8N4Q1 | protein import; calcium transport regulation |

**Table 1.** (*Continued.*)

| section | protein name | | gene | accession (NCBI; UniProt) | function(s) |
| --- | --- | --- | --- | --- | --- |
| | abbreviation | full name(s) | | | |
| immune signalling | MAVS | mitochondrial antiviral signalling protein | MAVS | 57506; Q7Z434 | immune signalling |
| | TRADD | TNFRSF1A associated via death domain | TRADD | 8717; Q15628 | immune signalling |
| | TRAF3 | TNF receptor-associated factor 3 | TRAF3 | 7187; Q13114 | immune signalling |
| | STING | transmembrane protein 173; stimulator of interferon genes | TMEM173 | 340061; Q86WV6 | immune signalling |
| | IRF3 | interferon regulatory factor 3 | IRF3 | 3661; Q14653 | immune signalling |
| | IRF7 | interferon regulatory factor 7 | IRF7 | 3665; Q92985 | immune signalling |
| | NLRX1 | NLR family member X1 | NLRX1 | 79671; Q86UT6 | immune signalling |
| | NLRP3 | NLR family pyrin domain containing 3 | NLRP3 | 114548; Q96P20 | immune signalling |
| | IL-1β | interleukin 1 beta | IL1B | 3553; P01584 | immune signalling |
| cell differentiation | Ras | K-Ras proto-oncogene, GTPase | KRAS | 3845; P01116 | cell proliferation |
| | Raf | Raf-1 proto-oncogene, serine/threonine kinase | RAF1 | 5894; P04049 | cell proliferation |
| | Pdk2 | pyruvate dehydrogenase kinase 2 | PDK2 | 5164; Q15119 | metabolism regulation |
| | Oct4 | POU class 5 homeobox | POU5F1 | 5460; Q01860 | stem cell differentiation |
| | Sox2 | SRY-box transcription factor 2 | SOX2 | 6657; P48431 | stem cell differentiation |
| | Nanog | Nanog homeobox | NANOG | 79923; Q9H9S0 | stem cell differentiation |
| | ZFP42 | ZFP42 zinc finger protein | ZFP42 | 132625; Q96MM3 | stem cell pluripotency |
| cell death | Bax | BCL2 associated X, apoptosis regulator | BAX | 581; Q07812 | intrinsic apoptosis |
| | Bak | BCL2 antagonist/killer1 | BAK1 | 578; Q16611 | intrinsic apoptosis |
| | Cyt c | cytochrome c, somatic | CYCS | 54205; P99999 | intrinsic apoptosis |
| | Diablo | Diablo IAP-binding mitochondrial protein | DIABLO | 56616; Q9NR28 | intrinsic apoptosis |
| | Htra2 | HtrA serine peptidase 2 | HTRA2 | 27429; O43464 | intrinsic apoptosis |
| | EndoG | endonuclease G | ENDOG | 2021; Q14249 | caspase-independent apoptosis |
| | AIF | apoptosis-inducing factor mitochondria associated 1 | AIFM1 | 9131; O95831 | caspase-independent apoptosis |
| | VDAC2 | voltage-dependent anion channel 2 | VDAC2 | 7417; P45880 | intrinsic apoptosis; ion permeability |

(*Continued.*)

royalsocietypublishing.org/journal/rsob  Open Biol. 9. 190126

**Table 1.** (Continued.)

| section | protein name | | full name(s) | accession (NCBI; UniProt) | gene | function(s) |
| | abbreviation | | | | | |
| --- | --- | --- | --- | --- | --- | --- |
| mitochondrial quality control | ATF5 | | activating transcription factor 5 | 22809; Q9Y2D1 | ATF5 | mitochondrial unfolded protein response |
| | ATF4 | | activating transcription factor 4 | 468; P18848 | ATF4 | integrated stress response |
| | hTim17a | | translocase of inner mitochondrial membrane 17A | 10440; Q99595 | TIMM17A | protein import; mitochondrial stress response |
| | MCL1 | | MCL1 apoptosis regulator, BCL2 family member; myeloid cell leukaemia 1 | 4170; Q07820 | MCL1 | intrinsic apoptosis |
| | PINK1 | | PTEN-induced kinase 1 | 65018; Q9BXM7 | PINK1 | mitophagy |
| | PARKIN | | Parkin RBR E3 ubiquitin protein ligase | 5071; O60260 | PRKN | mitophagy |
| | PARL | | presenilin-associated rhomboid like | 55486; Q9H300 | PARL | mitophagy |
| | hTom22 | | translocase of outer mitochondrial membrane 22 | 56993; Q9NS69 | TOMM22 | protein import; mitophagy |
| | FUNDC1 | | FUN14 domain containing 1 | 139341; Q8IVP5 | FUNDC1 | mitophagy |
| | BCL2L13 | | BCL2-like 13 | 23786; Q9BXK5 | BCL2L13 | mitophagy |

transcription of interleukins and pro-inflammatory cytokines [106–108] (figure 4a). Interestingly, MAVS dimers and many of its adaptors co-immunoprecipitate with hTom70 of the TOM complex, the overexpression of which increases the signalling response [109]. MAVS signalling is also affected by ROS and negatively regulated by Nlrx1, a binding partner of Complex III and MAVS [110,111] (figure 4a). As mitochondrial protein import and oxidative metabolism can be hijacked by virulence factors [112], these interactions may make MAVS sensitive to consequences of infection. Finally, if mitochondria are compromised by infection, the increased ROS and release of mtDNA into the cytosol can activate the NLRP3 inflammasome to evoke an inflammatory response [113,114] (figure 4a).

Mitochondrial metabolism also directs rapid changes to specialized immune cells during infection. Changes in membrane potential can activate or supress M2 macrophages [115,116] and M1 macrophages shunt intermediates from the TCA cycle to generate nitrous oxide, IL-1β and the antibacterial itaconic acid [117,118]. Furthermore, the phagocytic abilities of macrophages depend on mitochondrial ROS production to destroy internalized pathogens [119]. Naive T-cells display increases in mitochondrial mass, mtDNA copy number, glycolysis, and glutamine metabolism during differentiation for rapid proliferation and to escape quiescence [120,121]. Metabolic remodelling then also decides the T-cells' mature fate [122,123], by altering cristae architecture [124] or by direct effect of metabolites on epigenetic transcription regulation [125].

(Recommended further reading on mitochondrial immune signalling: [106,118,126].)

# 5. Cell cycle, differentiation and death

Mitochondria are implicitly tied to cell-cycle control as providers of energy and nucleotides; however, they also coordinate checkpoints and respond to signals of proliferation. To meet the metabolic demand of mitosis, mitochondrial mass and membrane potential increase from $G_1/S$ until late mitotic stage [127]. Indeed, hyperpolarization and increased ATP production inhibit AMP kinase to allow cyclinE-mediated entry to S-phase [128]. In the late G2 stage of dividing *S. cerevisiae*, the cyclinB1/Cdk1 complex traffics to mitochondria to phosphorylate Complex I subunits and Tom6, stimulating oxidative metabolism both directly and indirectly via increased protein import [129,130]. During mitosis, a highly fused and reticular mitochondrial network progressively fragments to small tubular organelles that segregate in anticipation of cytokinesis [127,131]. Mitochondria can also delay cell-cycle progression to increase their biogenesis [132], because of insufficient nucleotide production [133], or because of ROS accumulation [134]. Moreover, the fusion mediator Mfn2 can sequester both Ras and Raf to inhibit proliferative signalling [135].

Stem cell differentiation also relies on mitochondria as a 'metabolic switch'. Human embryonic stem cells are glycolytic; however, they develop mature cristae, rapidly replicate mtDNA and increase ATP production upon differentiation [136]. In haematopoietic stem cell differentiation, the downregulation of Pdk2, an inhibitor of pyruvate dehydrogenase, releases suppression of acetyl-coA production and enables oxidative phosphorylation [137]. The subsequent increase in ROS production and oxidative phosphorylation during differentiation drives upregulation of mitochondrial antioxidant proteins by the transcription factors Oct4, Sox2 and Nanog

**Figure 3.** Mitochondria coordinate essential metabolic processes. (*a*) Mitochondria are best known for housing the protein machinery required for generating ATP. When oxygen is available, most cells will generate ATP through oxidative phosphorylation, where electrons harvested through catabolic reactions are used to power ATP synthase. Electrons are obtained through the TCA cycle, which occurs in the matrix and consists of eight enzymatic reactions. Acetyl-CoA is the primary input for the TCA cycle, and can be obtained through metabolism of glucose, fatty acids and amino acids. Electrons extracted during the TCA cycle are loaded onto $NAD^+$ and $FAD^{2+}$. Electrons are subsequently transferred from NADH and $FADH_2$ onto Complexes I and II of the electron transport chain. Electrons are passed through Complexes III and IV, which transport protons into the intermembrane space. Protons are allowed to flow back into the matrix through ATP synthase (Complex V), which uses the energy of the proton gradient to convert ADP to ATP. (*b*) Mitochondrial one-carbon (1C) metabolism comprises a series of parallel and reversible reactions which occur in the cytosol and mitochondrial matrix. In proliferating cells, the reaction normally proceeds in a specific direction such that formate produced within mito-chondria can be used for biosynthetic processes in the cytosol. Within the mitochondria, THF and serine imported from the cytosol are acted upon sequentially by SHMT2, MTHFD2 and MTHFD1 L to produce formate, which is exported back into the cytosol. Cytosolic MTHFD1 loads formate onto THF to form charged folate intermediates that can be used to synthesize purine and pyrimidine nucleotides. Mitochondrial 1C metabolism is also an important source of glycine. (*c*) The mitochondrial matrix functions as an important storage site for calcium ions. Mitochondrial calcium uptake often occurs at ER contact sites, where large volumes of $Ca^{2+}$ can be released through IP3R3. Calcium can pass freely through the outer membrane via VDAC channels and is transported across the intermembrane space and inner membrane through the coordinated function of a MICU1/MICU2 dimer docking to MCU in the inner membrane. Calcium can exit the mitochondrial matrix through LETM1 or SLC8B1 (in exchange for $H^+$ or $Na^+$, respectively) and can cross the outer membrane through VDACs or NCX3.

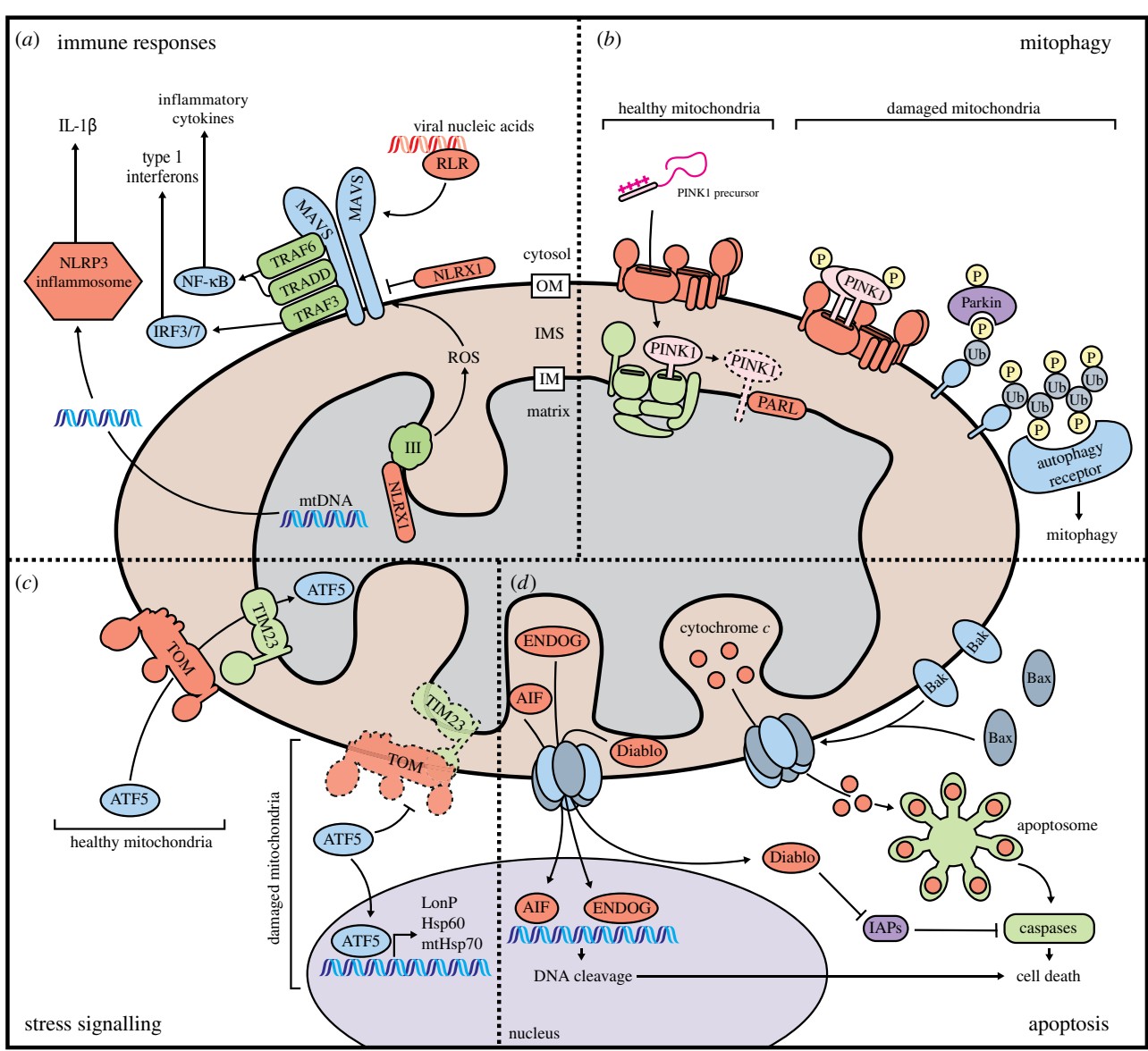

**Figure 4.** Mitochondria make crucial contributions to diverse cellular processes. (*a*) The mitochondrial outer membrane is the site of important signalling events during the innate immune response. Detection of viral nucleic acids by Rig-like receptors (RLRs) induces dimerization of MAVS, a protein of the mitochondrial outer membrane. Dimerized MAVS recruits signalling adaptors that initiate downstream activation of IRF3/7 and NF-κB, transcription factors that induce expression of type I interferons and pro-inflammatory cytokines. MAVS is regulated by NLRX1, a protein which downregulates MAVS when localized to the outer membrane, but activates MAVS when at the inner membrane by interacting with Complex III to induce ROS production. Release of mtDNA during infection can also activate the NLRP3 inflammasome. (*b*) Mitophagy is a process that allows damaged mitochondria to be identified and destroyed. Under normal conditions, PINK1 is imported into mitochondria and degraded by PARL. When mitochondria are damaged, import is impaired and PINK1 accumulates in the TOM complex at the outer membrane. Autophosphorylated and active PINK1 at the outer membrane phosphorylates monoubiquitin molecules on outer membrane proteins, recruiting and activating the E3 ubiquitin ligase Parkin. Activated Parkin synthesizes polyubiquitin chains that recruit autophagy receptors to initiate mitophagy. (*c*) Mitochondrial proteostatic stress is sensed through the partitioning of the transcription factor ATF5 between the mitochondria and the nucleus. Under normal conditions, ATF5 is imported into and sequestered within mitochondria. If mitochondrial protein import becomes compromised, ATF5 is trafficked into the nucleus, where it upregulates expression of genes that enhance proteostasis. (*d*) Mitochondria play crucial roles in the initiation of apoptosis. In response to pro-apoptotic stimuli, Bax and Bak oligomerize in the outer membrane to form pores that allow for efflux of apoptogenic proteins (Cytochrome *c*, Diablo, AIF and Endonuclease G) from the intermembrane space into the cytosol. Cytochrome *c* binds to Apaf-1 to induce formation of the apoptosome and activation of caspases. Diablo blocks inhibitors of apoptosis (IAPs) which would otherwise mitigate the effect of caspases. AIF and Endonuclease G translocate into the nucleus where they contribute to destruction of the genome.

[138]. Mitochondrial fusion is believed to facilitate these metabolic changes, although the importance of specific proteins and fission/fusion balance may be cell-type specific [139–141]. This is supported by somatic cell reprogramming studies showing deletion of Mfn2 permits pluripotency as glycolysis becomes predominant over oxidative phosphorylation [142]; the same effect being achieved by the pluripotency factor ZFP42 activation of Drp1 [143].

If cellular conditions or external insults are too harsh, mitochondria can trigger multiple forms of cell death.

Apoptosis, or programmed cell death, can be elicited from extrinsic signalling via the Fas, TRAIL and TNFα receptors or intrinsic insults such as DNA damage, Ca²⁺ overload, ROS and ER stress [144]. Mitochondria contribute to the extrinsic pathway but are the nexus of the intrinsic apoptotic pathway. In the latter pathway, cytosolic pro-apoptotic Bax oligomerizes with Bak at the outer membrane to permeabilize mitochondria and release pro-apoptotic proteins, including cytochrome *c*, Diablo, Htra2, Endonuclease G and AIF (figure 4*d*) [145]. In the cytosol, cytochrome *c* nucleates

royalsocietypublishing.org/journal/rsob Open Biol. 9: 190126

the formation of the apoptosome and activation of the caspases that dismantle the cell in an immunologically silent manner. Cytosolic Diablo and Htra2 block inhibitors of caspase activation, which would otherwise protect the cell from basal cytochrome c leakage [146,147]. Endonuclease G and AIF translocate to the nucleus to fragment DNA (figure 4d), AIF first requiring proteolytic cleavage of its transmembrane domain [148–150]. AIF is normally part of the intermembrane space import machinery, or MIA complex, anchoring the oxidoreductase hMia40 to the inner membrane. The outer membrane protein VDAC2 protects against apoptosis by sequestering Bak [151,152], yet new evidence suggests it may be required for Bax-mediated apoptosis [153]. Emerging research also implicates mitochondria in alternate and less-studied cell-death pathways such as ROS-induced necrosis [154], immune-activated necroptosis [155], ferroptosis [156,157] and parthanotosis [158].

(Recommended further reading on mitochondria in the cell cycle: [159,160]; on differentiation [161–163]; on cell death: [164,165].)

# 6. Mitochondrial quality control

The loss of mitochondrial function has profound negative effects on cellular health; therefore, multiple quality control and stress response mechanisms have evolved. The mitochondrial unfolded protein response (mtUPR) detects proteostatic stress within mitochondria [166]. Central to the mtUPR is the transcription factor ATF5. When stress causes protein import and/or electron transport chain dysfunction ATF5 accumulates in the nucleus to transcribe mitochondrial chaperones and protease genes (figure 4c) [167,168]. The *Caenorhabditis elegans* homologue ATFS-1 has also been shown to repress translation of the electron transport chain subunit and assembly proteins from both mitochondrial and nuclear genomes [169]. Translation of ATF5 is partly controlled by its homologue ATF4, both of which are upregulated in the integrated stress response (ISR) [170,171]. The ISR can be triggered by ER stress, amino acid starvation or degradation of hTim17A, a TIM23 complex subunit [172,173]. The ISR is characterized by phosphorylation of eIF2α, leading to global reduction of translation and selective induction of cytoprotective genes including pro-survival MCL1 and autophagy proteins. This illustrates the preference for clearance of defective organelles over controlled cell death although the response may alter with cell type or insult [174].

The selective autophagic clearance of mitochondria is termed mitophagy and is controlled by the mitochondrial serine/threonine protein kinase PINK1 and the E3 ubiquitin ligase Parkin. PINK1 is constitutively imported into healthy mitochondria through the TOM complex and laterally released into the inner membrane by TIM23 [175] before cleavage by the PARL protease (figure 4b) [176]. Depolarization of the inner membrane in defective mitochondria prevents import of PINK1, causing it to oligomerize at the outer membrane TOM complex [177], where it becomes auto-phosphorylated [178]. This triggers phospho-PINK1 phosphorylation of basal outer membrane monoubiquitin and recruits Parkin to rapidly poly-ubiquitinate outer membrane proteins for the recruitment of autophagosome factors (figure 4b) [179,180]. Recent data suggest that mitochondria can identify and initiate mitophagy of specific tubules [181], while mitophagy induced by CSNK2/

CK2 phosphorylation of hTom22, FUNDC1 and BCL2L13 suggests a potential cytoplasmic influence or pathway [182–185]. Additionally, observations of transcellular mitophagy in astrocytes illustrate much is still unknown in these processes [186].

New stress responses are emerging that demonstrate the reciprocal communication between mitochondria and cytoplasm. Ablation of MIA import pathways in *S. cerevisiae* activates the proteasome to mitigate mitochondrial precursor accumulation in the cytosol [187]. This correlates with the mammalian, intermembrane space-specific mtUPR (mtUPR$_{IMS}$) where ERRα transcriptional activity upregulates intermembrane space proteases and activates the proteasome [188,189]; the proteasome being previously shown to degrade unfolded intermembrane space proteins that retrotranslocate to the cytosol [190]. In *S. cerevisiae*, the proteasome is also engaged by Ubx2 to clear mitochondrial protein precursors arrested during translocation, blocking the TOM complex [191]. Reciprocally, mitochondria can degrade defective proteins to aid cytosolic proteostasis. In *S. cerevisiae*, cytosolic Vms1 can remove mistranslated mitochondrial precursors from stalled ribosomes and direct their import for intra-mitochondrial degradation [192] and aggregation-prone cytosolic proteins may be imported for intra-mitochondrial degradation if cytosolic Hsp70s fail [193]. Intriguing for further research are reports of lysosomal fusion of mitochondria-derived vesicles enriched for non-natively oxidized proteins [194,195] and the extracellular jettison of aggregates by neurons of *C. elegans* [196].

(Recommended further reading on mitochondrial quality control: [197–199]; on mitophagy: [200,201].)

# 7. Concluding remarks

This review illustrates the importance of mitochondria to eukaryotic cellular functions. As mitochondrial biologists we are frequently surprised by novel pathways or protein networks that involve mitochondria and/or mitochondrial proteins. Mitochondrial protein import and structural dynamics provide the means for rapid alterations in activity to facilitate biological responses to signalling molecules, nutrient availability and pathogenic insult. The temporal coordination of mitochondrial energetics and their biosynthetic capacity drives cell proliferation and differentiation. However, the highly reactive biochemistry compartmentalized in the organelle makes it capable of inducing cell death and necessitates quality control mechanisms. An understanding of this interplay between mitochondrial functions and their diverse cellular implications is therefore critical to a comprehensive holistic model of cellular homeostasis and biochemistry. The importance of this is evident in the escalating occurrence of mitochondria in post-genomic medical research [202]. Although mitochondria are undeniably hubs of cellular biochemistry, further fundamental research is required. In particular, elucidating how the mitochondrion regulates and integrates the various pathways it is associated with, in specialized cells/tissue types and in the context of health and in disease, will help uncover the true depth of influence this amazing organelle has on eukaryotic cells.

Data accessibility. This article does not contain any additional data.
Competing interests. We declare we have no competing interests.
Funding. We received no funding for this study.

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
