## [Reviewer comments · Open Biology]

Review History

RSOB-19-0126.R0 (Original submission)

Review form: Reviewer 1

Recommendation

Accept with minor revision (please list in comments)

Scientific importance: Is the manuscript an original and important contribution to its field?

Excellent

General interest: Is the paper of sufficient general interest?

Excellent

Quality of the paper: Is the overall quality of the paper suitable?

Excellent

It is a condition of publication that authors make their supporting data, code and materials available - either as supplementary material or hosted in an external repository. Please rate, if applicable, the supporting data on the following criteria.

Is it accessible?

N/A

Is it clear?

N/A

Is it adequate?

N/A

Do you have any ethical concerns with this paper?

No

Comments to the Author

This is an excellent, up-to-date review on mitochondrial biology, dynamics and regulation. The authors discuss the large variety of mitochondrial functions and their integration into a dynamic network. The chapters are structured very well and are presented in a clear and logic arrangement. The figures are outstanding. I fully recommend publication of this excellent review article.

Minor points:

- Explain abbreviations, e.g. DRP1, MFF, MiD49, MiD51. Due to the large number of factors presented, it would be helpful for the reader to include a table where all abbreviations are explained (possibly arranged in functional groups such that the table of abbreviations also provides an overview of the numerous mitochondrial tasks).
- Indicate that HUMAN mitochondria are meant in the part describing that mtDNA encodes 13 polypeptides.
- TIM23 can also mediate the import of polytopic (multi-spanning) inner membrane proteins, e.g. ABC transporters. To distinguish the functions of TIM23 and TIM22, the authors can include that TIM22 mediates the insertion of NON-CLEAVABLE polytopic membrane proteins.

Decision letter (RSOB-19-0126.R0)

05-Jul-2019

Dear Dr Stojanovski

We are pleased to inform you that your manuscript RSOB-19-0126 entitled "Mitochondria - hubs for regulating cellular biochemistry: emerging concepts and networks" has been accepted by the Editor for publication in Open Biology. The reviewer(s) have recommended publication, but also suggest some minor revisions to your manuscript. Therefore, we invite you to respond to the reviewer(s)' comments and revise your manuscript.

Please submit the revised version of your manuscript within 14 days. If you do not think you will be able to meet this date please let us know immediately and we can extend this deadline for you.

- 1) A text file of the manuscript (doc, txt, rtf or tex), including the references, tables (including captions) and figure captions. Please remove any tracked changes from the text before submission. PDF files are not an accepted format for the "Main Document".
- 2) A separate electronic file of each figure (tiff, EPS or print-quality PDF preferred). The format should be produced directly from original creation package, or original software format. Please note that PowerPoint files are not accepted.
- 3) Electronic supplementary material: this should be contained in a separate file from the main text and meet our ESM criteria (see <http://royalsocietypublishing.org/instructions-authors#question5>). All supplementary materials accompanying an accepted article will be treated as in their final form. They will be published alongside the paper on the journal website and posted on the online figshare repository. Files on figshare will be made available approximately one week before the accompanying article so that the supplementary material can be attributed a unique DOI.

Online supplementary material will also carry the title and description provided during submission, so please ensure these are accurate and informative. Note that the Royal Society will not edit or typeset supplementary material and it will be hosted as provided. Please ensure that the supplementary material includes the paper details (authors, title, journal name, article DOI). Your article DOI will be 10.1098/rsob.2016[last 4 digits of e.g. 10.1098/rsob.20160049].

- 4) A media summary: a short non-technical summary (up to 100 words) of the key findings/importance of your manuscript. Please try to write in simple English, avoid jargon, explain the importance of the topic, outline the main implications and describe why this topic is newsworthy.

Images

Data-Sharing

It is a condition of publication that data supporting your paper are made available. Data should be made available either in the electronic supplementary material or through an appropriate

repository. Details of how to access data should be included in your paper. Please see <http://royalsocietypublishing.org/site/authors/policy.xhtml#question6> for more details.

Data accessibility section

Sincerely,
The Open Biology Team
<mailto:openbiology@royalsociety.org>

Reviewer(s)' Comments to Author:

Referee: 1

Comments to the Author(s)

This is an excellent, up-to-date review on mitochondrial biology, dynamics and regulation. The authors discuss the large variety of mitochondrial functions and their integration into a dynamic network. The chapters are structured very well and are presented in a clear and logic arrangement. The figures are outstanding. I fully recommend publication of this excellent review article.

Minor points:

- Explain abbreviations, e.g. DRP1, MFF, MiD49, MiD51. Due to the large number of factors presented, it would be helpful for the reader to include a table where all abbreviations are explained (possibly arranged in functional groups such that the table of abbreviations also provides an overview of the numerous mitochondrial tasks).
- Indicate that HUMAN mitochondria are meant in the part describing that mtDNA encodes 13 polypeptides.
- TIM23 can also mediate the import of polytopic (multi-spanning) inner membrane proteins, e.g. ABC transporters. To distinguish the functions of TIM23 and TIM22, the authors can include that TIM22 mediates the insertion of NON-CLEAVABLE polytopic membrane proteins.

Decision letter (RSOB-19-0126.R1)

12-Jul-2019

Dear Dr Stojanovski

We are pleased to inform you that your manuscript entitled "Mitochondria - hubs for regulating cellular biochemistry: emerging concepts and networks" has been accepted by the Editor for publication in Open Biology.

Sincerely,

The Open Biology Team
mailto: openbiology@royalsociety.org